# Patients with Systemic Juvenile Idiopathic Arthritis (SJIA) Show Differences in Autoantibody Signatures Based on Disease Activity

**DOI:** 10.3390/biom13091392

**Published:** 2023-09-15

**Authors:** Julie Krainer, Michaela Hendling, Sandra Siebenhandl, Sabrina Fuehner, Christoph Kessel, Emely Verweyen, Klemens Vierlinger, Dirk Foell, Silvia Schönthaler, Andreas Weinhäusel

**Affiliations:** 1Center for Health and Bioresources, Molecular Diagnostics, AIT Austrian Institute of Technology GmbH, Giefinggasse 4, 1210 Vienna, Austria; michaela.hendling@ait.ac.at (M.H.); klemens.vierlinger@ait.ac.at (K.V.); silvia.schoenthaler@ait.ac.at (S.S.); 2Pediatric Rheumatology & Immunology, University Children’s Hospital, 48149 Münster, Germany; sabrina.fuehner@uni-muenster.de (S.F.); kesselc@uni-muenster.de (C.K.); e_verw01@uni-muenster.de (E.V.); dirk.foell@ukmuenster.de (D.F.)

**Keywords:** systemic juvenile idiopathic arthritis, systemic autoinflammatory disease, protein microarray, autoantibodies

## Abstract

Systemic juvenile idiopathic arthritis (SJIA) is a severe rheumatic disease in children. It is a subgroup of juvenile idiopathic arthritis (JIA; MIM #604302), which is the most common rheumatic disease in children. The diagnosis of SJIA often comes with a significant delay, and the classification between autoinflammatory and autoimmune disease is still discussed. In this study, we analyzed the immunological responses of patients with SJIA, using human proteome arrays presenting immobilized recombinantly expressed human proteins, to analyze the involvement of autoantibodies in SJIA. Results from group comparisons show several differentially reactive antigens involved in inflammatory processes. Intriguingly, many of the identified antigens had a high reactivity against proteins involved in the NF-κB pathway, and it is also notable that many of the detected DIRAGs are described as dysregulated in rheumatoid arthritis. Our data highlight novel proteins and pathways potentially dysregulated in SJIA and offer a unique approach to unraveling the underlying disease pathogenesis in this chronic arthropathy.

## 1. Introduction

Systemic juvenile idiopathic arthritis (SJIA) is a severe rheumatic disease in children. It is a subgroup of juvenile idiopathic arthritis (JIA; MIM #604302), which is the most common rheumatic disease in children [1,2]. JIA is seen as an autoimmune disease, whereas SJIA has characteristics of an autoinflammatory disease [3]. Based on the International League of Associations for Rheumatology (ILAR), the clinical definition of patients with SJIA includes arthritis in one or more joints for at least 6 weeks with or preceded by fever for at least three consecutive days and accompanied by one or more of the following: evanescent (nonfixed) erythematous rash, generalized lymph node enlargement, hepatomegaly and/or splenomegaly, and serositis [2]. A life-threatening complication is Macrophage Activation Syndrome (MAS), affecting about 10% of children with SJIA, and subclinically up to 30–40% [4,5,6] with a reported mortality rate of 8–22% [1,5,7]. Even though SJIA was first described in 1897 by George F Still, the pathophysiology is still not very well understood.

The diagnosis of SJIA often comes with a significant delay, as fever of unknown origin can be a symptom for a broad spectrum of diseases like infection [8]. Additionally, the characteristic sign of arthritis in patients with SJIA is often preceded by fever, and the initial presentation of systemic inflammation does not differ from infection by clinical parameters like elevated acute phase reactants or high ferritin values [9,10]. Currently the criteria for SJIA diagnosis are based on clinical features, lacking laboratory biomarker. There are a few diagnostic markers proposed for the diagnosis of SJIA patients. Reported SJIA biomarkers are IL6, IL18, S100A8/A9, S100A12, and OSM [9,11,12,13]. Results show that first-line treatments like with Anakinra, a recombinant Interleukin-1 receptor antagonist (IL-1RA), are very successful to prevent chronic, destructive synovitis [14,15]. Apart from the first-line treatment, other therapeutics include IL-1-targeted biologics like Canakinumab (anti-interleukin-1β monoclonal antibody), Rilonacept (fusion protein IL-1 receptor), and IL-6 receptor monoclonal antibody Tocilizumab [16,17,18,19,20]. Other treatments for symptom management are nonsteroidal anti-inflammatory drugs (NSAIDs) and glucocorticosteroids with disease-modifying drugs [17,21].

The classification of SJIA as an autoinflammatory or autoimmune disease has been discussed in the literature, as the disease also shows signs of both disease groups [22,23,24,25]. In 2006, McGonagle et al. proposed a disease continuum model combining the adaptive and the innate immune system with their respective disease group [26]. Following the disease continuum model, between autoinflammatory and autoimmune diseases, SJIA falls in between as a multifactorial SAIDs having characteristics from both disease groups. Autoinflammatory features of SJIA include the recurrent fevers, as seen in classic fever syndromes, whereas destructive arthritis when untreated is an autoimmune disease feature [10,22].

A characteristic feature of autoimmune diseases is the presence of autoantibodies, and following the historic definition, autoinflammatory disease are known to lack those. In autoimmune diseases, autoantibodies are used for early diagnosis and classification [27]. There are several studies analyzing autoantibodies in SJIA patients. Huegle et al. [24] showed increased serum antinuclear antibodies and positive rheumatoid factor in SJIA patients over time. Another study by Bloom et al. [28] showed a prevalence of anti-endothelial cell antibodies in children with SJIA compared to healthy controls. Other studies analyzing antiphospholipid antibodies [29] or thyroid-specific antibodies, celiac-specific antibodies, and antibodies for connective tissue diseases [30] in SJIA patients could not show a distinct difference compared to the general population. The present study is the first study known to the authors that goes beyond analyzing single antibody responses but identifies multivariate autoantibody signatures from a highly parallel (16k) screening assay.

In this study, we analyzed the immunological responses of patients with SJIA, using human proteome arrays presenting immobilized recombinantly expressed human proteins. These arrays have been previously used in studies analyzing antibody profiles in cancer patients and patients with rheumatic arthritis and ulcerative colitis [31,32,33,34].

## 2. Materials and Methods

### 2.1. Samples

For this study, we analyzed the autoantibody reactivity for 66 patients with systemic juvenile idiopathic arthritis and 23 controls described in Table 1. The study groups included 26 samples with active SJIA, 40 patients with inactive SJIA, and 23 healthy controls. For 18 patients, paired samples during active and inactive disease were taken. The definition for active SJIA patients is elevated CRP, elevated ESR and a physician global assessment > 1. Some healthy controls were also used as an internal control on multiple microarrays on different analysis days. In total, 110 serum samples were analyzed. Healthy controls were recruited at the University of Muenster in Germany; patient samples are from the German multicenter national registry for Autoinflammatory Syndromes in Children (AID-Net). At the time of sampling, the SJIA patients had a median age of 11.5 years (range: 1.0–20.8 years), the age of diagnosis was 7.1 years (range: 0.6–17.4 years), and the disease duration was 3.5 years (range: <0.1–17.3); 47% (31 of 66) were female. Laboratory findings show that active SJIA patients had a median C-reactive protein (CRP) of 6.8 mg/dL and the inactive SJIA group had a median C-reactive protein of <0.5 mg/dL; the Erythrocyte Sedimentation Rate (ESR) for active SJIA patients was 5.4 mm/hour, while it was 4.2 mm/hour for inactive SJIA patients. The leukocytes were at 15.15/nL in active SJIA patients and 5.9/nL in inactive SJIA patients. The S100 calcium binding protein A12 (S100A12) was at 1195 ng/mL for active and 39 ng/mL for inactive SJIA patients, and the S100 calcium binding protein A8/A9 (S100A8/A9) was at 7320 ng/mL for active patients and 560 for inactive SJIA patients. Treatment consisted of steroids, biological disease modifying anti-rheumatic drugs (bDMARDs), and conventional disease modifying anti-rheumatic drugs (csDMARDs). Overall, 47.8% of the healthy controls (n = 23) were female and had a median age of 14.7 years. All healthy controls had a CPR < 0.5 mg/dL, a median of 3 mm/hour for ESR, and the leukocytes were at 5.5/nL (see Table 1). Appendix A describes the symptoms of the patients. All parents/patients signed a written consent form before entering the study. To collect serum, whole blood from patients and controls was centrifuged at a max. of 4000 rpm at room temperature for up to 10 min within 4 h of withdrawal. The serum was transferred to a new tube and immediately stored at −80 °C. All samples were stored at −80 °C until the experiments were conducted.

### 2.2. IgG Purification

Immunoglobulin G (IgG) was purified using the Melon Gel IgG Purification Spin Plate Kit (#45208, Thermo Scientific ^TM^, Waltham, MA, USA 02451) following the manufacturers’ protocol. In short, after equilibration of Melon Gel 96-well spin plate and purification buffer to room temperature, the storage buffer is removed by centrifuging for 1 min at 1000× *g*. Then, 15 μL of serum is diluted in 95 μL of Melon Gel Purification Buffer and transferred to the 96-well spin plate. The plate assembly is then placed on a plate shaker at room temperature and incubated for 5 min. To collect the purified IgG antibodies, the plate is centrifuged at 1000× *g* for 1 min.

Quantifications of IgG antibody concentration were performed by A280 UV-spectrophotometry using the Epoch Multi-Volume Spectrophotometer System (BioTek Instruments, Winooski, VT, USA). Using the resulting concentrations, the purified IgG was diluted with Melon Gel Purification Buffer (#89972, Thermo Fisher Scientific) to a final concentration of 0.2 mg/mL in 250 µL total volume. To check IgG purity, random samples were run on a pre-casted SDS-PAGE using NuPAGE^®^ Novex 4–12% Bis-Tris Gels (Life Technologies, Carlsbad, CA, USA). The integrity of all checked samples was confirmed.

### 2.3. Protein Microarray Processing

For this study, 110 AIT’s-16k protein microarray slides were processed using purified IgG samples of patients with systemic juvenile idiopathic arthritis and controls (see Table 1). Due to handling capacity, the slides were processed in three runs on different days.

The in-house printed AIT’s-16k protein microarray comprises 15312 full-length *E. coli* cDNA expression clones. In total, 6124 recombinantly expressed proteins, acting as antigens, are covered and annotated with gene symbols. Each recombinant protein is represented by several expression clones and derived from human fetal brain, T-cells, lung, and colon protein expression libraries from the UniPEx–human in-frame cDNA protein expression library (engine, Berlin, Germany, previously ImaGenes, Berlin, Germany, previously RZPD Ressourcenzentrum für Genomforschung, Germany).

The protein expression, purification, and production of the used protein microarrays were performed as previously described [31,32,35,36]. Proteins, negative controls (elution buffer) and positive controls (*E. coli* lysate) were printed in duplicates onto SU8 epoxy-dip-coated glass slides using the contact a NanoPrint™LM210 device.

To process the microarray slides, they were first incubated in a slide rack for 10 min in 70 °C preheated 2% SDS solution and blocked immediately for 30 min with DIG Easy Hyb blocking solution (#11603558001 Roche, Basel, Swiss) at room temperature. Meanwhile, the final hybridization mix is prepared by adding 250 µl of dilution solution (2× PBS, 0.2% TritonX-100, 6% milk powder) to each prepared 250 μL IgG sample. The blocked slides are then washed with wash buffer (1× PBS, 0.1% TritonX-100) for 5 min. This washing step was repeated twice using fresh wash buffer and discarding old wash buffer after each repetition. Lastly, the slides were washed in ddH2O for 10 s and dried in a swinging bucket rotor centrifuge for 5 min at 400× *g*. After 490 μL of the final hybridization mix was applied, the slides were hybridized for 4 h with constant rotation (12 rpm) at room temperature in an Agilent rotating oven (Agilent, Santa Clara, CA, USA). Afterwards, the slides were rinsed three times with wash buffer (1× PBS, 0.1% TritonX-100) for 5 min, finally washed with ddH2O for 10 s, and dried in a swinging bucket rotor centrifuge for 5 min at 400× *g*. The slides are then placed in detection solution (1:10,000 diluted Alexa Fluor 647 goat anti-human IgG (H+L) (Life Technologies-Thermo Fisher Scientific, Vienna, Austria) in 1× PBS, 0.1% Triton X 100, 3% milk powder) on a magnetic stirrer and incubated for 1 h in dark while stirring (300 rpm). After the slides were washed again 3 times in fresh PBST, rinsed in ddH2O for 10 s, and dried in a swinging bucket rotor centrifuge for 5 min at 400× *g*, the slides were scanned using an Agilent Microarray Scanner (settings: 10 μm resolution, TIFF: 16-bit, Red PMT: 90%, Green PMT: 1%, XDR: <NoXDR>).

After microarray processing and scanning, all slides were manually gridded using the Software GenePix^®^ Pro 6.0 (Molecular Devices, Sunnyvale, CA, USA). The resulting median fluorescence intensities per spot were calculated by subtracting the local median background. Due to handling capacity, the samples were processed in three runs on different days.

### 2.4. Protein Microarray Data Analysis

The analysis was conducted using R (Version 4.2). Data preprocessing and detection of differentially reactive antigens (DIRAG) were conducted using the R package limma. The data are preprocessed by background correction, normalization of microarray expression intensities, and averaging replicate spots. To identify the most optimal normalization method, hierarchical clustering was performed using three different normalization methods (no normalization, median normalization, and quantile normalization) on paired samples, as the intra-individual differences are smaller than the inter-individual difference. In total, 18 paired samples were analyzed, run 1 included 4 pairs, and run 3 included 14 pairs. Without normalization, 7 out of 14 pairs cluster with each other, quantile normalization resulted in 14 pairs clustering, and median normalization resulted in 12 pairs clustering (see Appendix A). Based on these results, quantile normalization was chosen as the optimal normalization method.

Due to handling capacity, the samples were processed on three analysis days. PCA analysis showed a separation between microarray intensity from run 1 and run 2 + 3 representing two different microarray printing batches; after batch correction using a parametric empirical Bayes adjustment (ComBat) [37] from the sva package [38], this effect was no longer visible (see Appendix A). As another quality control step, the same healthy controls were analyzed on multiple analysis runs. Without batch correction, 5 out of 21 healthy controls analyzed on multiple analysis runs clustered together during hierarchical clustering. After batch correction, 11 healthy controls clustered together; for two healthy controls, analyzed on all three analysis runs, two of three clustered together, showing the effect of the batch correction step (see Appendix A). Duplicate healthy controls were removed prior to group comparison.

To detect differentially reactive antigens (DIRAG), a moderated *t*-test based on linear models, taking the variance of all selected features into account, using the R package limma was performed. For the comparison of paired SJIA active vs. SJIA inactive, a moderated paired *t*-test was calculated. IgG concentrations were compared using a Welch two-sample *t*-test and a paired *t*-test. Gene set enrichment analysis of Reactome pathways [39] was performed with the resulting DIRAGs and their log2FC as a metric, using the R wrapper for the WEB-based GEne SeT AnaLysis Toolkit (WebGestalt) [40]. During GSEA, significant DIRAGs are ranked based on their log2FC. The resulting measure is the enrichment score, which is positive for overrepresented pathways in the first testing group or negative for overrepresented pathways in the second testing group. It represents if the DIRAGs of the ranked list are on the top (highest log2FC) or the bottom (lowest log2FC). All 6124 expressed proteins were set as background for GSEA. To adjust for multiple testing, *p*-values were corrected using the Benjamini–Hochberg correction [41]. Significance was set to *p* ≤ 0.05 throughout the analysis. Visual presentation of the data was completed using the R package ggplot2 [42]. Four group comparisons were calculated: (i) paired + unpaired active versus inactive SJIA, (ii) SJIA versus controls, and (iii) inactive versus controls.

## 3. Results

To identify DIRAG resulting from the IgG profiling of SJIA patients, we processed and analyzed 110 protein microarrays. The serum samples used in this study were collected at the University of Münster and through the AID-Net project. Regarding the experimental setup, batch correction using ComBat was performed for experiments run on three different analysis days.

For microarray experiments, the intra-individual difference is smaller than the inter-individual difference; hence, paired samples with active and inactive SJIA cluster together during hierarchical clustering. To identify the most optimal normalization method, no normalization, quantile normalization, and median normalization were compared using paired samples. Based on that, quantile normalization was chosen (details are described in the Methods section).

To compare immunomic differences, active and inactive disease states were compared to healthy controls separately.

### 3.1. IgG Concentration in Sera

To test for differences in IgG concentration isolated from serum between the analyzed groups, the concentrations of the disease groups were compared to the control group (see Figure 1A). The median IgG concentration upon MelonGel purification was 7.9 mg/mL (range: 2.26–24.99) and is within the expected range [43,44]. Looking at the three analysis days separately, analysis day 3 had a higher concentration (median 9.64 mg/mL), whereas concentrations on day 1 (median 6.43 mg/mL) and day 2 (median 7.1 mg/mL) were similar. Comparing the IgG concentrations of active and inactive SJIA to healthy controls shows no significant difference for each of the three analysis days. Differences in IgG concentration of paired SJIA samples were significant (*p* = 0.02) with a difference in mean of 2.5 mg/mL, where active SJIA samples have a higher concentration (mean = 1.36 mg/mL) than inactive SJIA samples (mean = 1.02 mg/mL) (see Figure 1B).

### 3.2. Differentially Reactive Antigen Analysis and Functional Enrichment

We applied a moderated *t*-test to detect differentially reactive autoantibodies between the analyzed groups: (i) SJIA active versus inactive, paired + unpaired, (ii) SJIA inactive versus controls, (iii) SJIA active versus controls. Reactome GSEA analysis was performed using significant DIRAGs resulting from group comparison. A volcano plot of the resulting DIRAGs for the group comparisons is shown in Figure 2.

#### 3.2.1. Active SJIA versus Inactive SJIA

Comparing the immunomic profile of 18 patients with paired active and inactive samples showed 1111 significant DIRAG (*p* ≤ 0.05), where 562 were higher reactive in active SJIA and 549 less reactive in active disease (see c in Table 2; for a full list, see Appendix A). The 1111 DIRAG covers 995 unique proteins. The log2FC ranges from 0.75 (isoleucyl-tRNA synthetase 2, mitochondrial (IARS2)) up to −0.91 (phospholipase C gamma 1 (PLCG1)). The top 5 proteins that are most reactive in active disease are IARS2 (log2FC = 0.75; *p* = 0.003), calmodulin 3 (CALM3; log2FC = 0.59; *p* = 0.0007), Meis homeobox 3 (MEIS3; log2FC = 0.58; *p* = 0.02), zinc finger protein 232 (ZNF232; log2FC = 0.54; *p* = 0.03), and MAF bZip transcription factor F (MAFF; log2FC = 0.535; *p* = 0.003). The top 5 DIRAG that are most reactive in inactive disease are three proteins of PLCG1 (log2FC = −0.91/−0.69/−0.67; *p* = 0.03/0.03/0.04), ArfGAP with GTPase domain, ankyrin repeat and PH domain 2 (AGAP2; log2FC = −0.81; *p* = 0.002), and amyloid beta precursor protein binding family A member 2 (APBA2; log2FC = −0.79; *p* = 0.009) (see c in Table 2, Figure 3, Appendix A). PLCG1 is covered by eight clones on the protein microarray. Of those, five are significantly less reactive in active SJIA patients, two proteins are less reactive in active SJIA patients (not significant), and one is more reactive in active SJIA patients (not significant).

To identify the enriched biological pathway, a gene set enrichment analysis was performed using the resulting DIRAGs from group comparison as input. The 1111 significant DIRAGs resulted in 20 significant pathways (FDR < 0.05) that are all enriched in inactive disease. To find the top gene sets while maximizing gene coverage, the weighted set cover method was used, which resulted in six pathway groups, covering 19 significant pathways. The six pathway groups are “Metabolism”, “Regulation of expression of SLITs and ROBOs”, “Infectious disease”, “Nonsense-Mediated Decay (NMD)," “Translation”, and “Transcriptional regulation by RUNX2”. The normalized enrichment score of the six pathway groups ranges from −2.4 (Metabolism) to −0.45 (Transcriptional regulation by RUNX2). Both in “Regulation of expression of SLITs and ROBOs” and “Nonsense-Mediated Decay (NMD)”, the list of DIRAGs include 68.6% of the total proteins within the respective pathway. In the highest ranked pathway “Metabolism” (NES = −2.4; FDR = 0.007), 102 out of 165 proteins are significantly higher reactive in inactive SJIA. The results of the GSEA can be found in the Appendix A.

Antigenic reactivities of unpaired SJIA samples (26 active SJIA, 40 inactive SJIA) have been analyzed, resulting in 282 significant (*p* ≤ 0.05; not adjusted) DIRAGs. Of those, 100 are more reactive in active disease, and 182 are more reactive in inactive disease. The top 5 proteins most reactive in active disease are FKBP prolyl isomerase like (FKBPL; log2FC = 0.87; *p* = 0.047), ADP ribosylation factor-interacting protein 2 (ARFIP2; log2FC = 0.67; *p* = 0.03), MAFF (log2FC = 0.55; *p* = 0.01), calmodulin 3 (CALM3; log2FC = 0.5; *p* = 0.001), and transmembrane protein 183A (TMEM183A; log2FC = 0.47; *p* = 0.03). Two proteins of PLCG1 are more reactive in inactive disease (log2FC = −0.58/−0.49; *p* = 0.03/0.01), AGAP2 (log2FC = −0.57; *p* = 0.04), claudin 3 (CLDN3; log2FC = −0.49; *p* = 0.01), and lamin A/C (LMNA; log2FC = −0.42; *p* = 0.002) (see d in Table 2, Figure 3; for a full list, see Appendix A).

GSEA with unpaired SJIA samples resulted in two significantly enriched pathways: “Transport of small molecules” is enriched in active SJIA (NES = 1.96; *p* = 0.002), and “Disease of signal transduction” is enriched in inactive disease (NES = −1.96; *p* = 0.003). Not significant but enriched pathways include “Innate Immune System” enriched in inactive disease (NES = −1.2) and “Adaptive Immune System” enriched in active disease (NES = 0.9) (see Appendix A).

Comparing paired and unpaired analysis of active versus inactive SJIA results in an overlap of 80 DIRAGs significant (not adjusted) in both comparisons. This overlap is significantly greater than the expected number of overlaps following the hypergeometric probability (*p* = 1.8 × 10^−27^). Myosin heavy chain 9 (MYH9) and PLCG1 are covered by two proteins each, which are all more reactive in inactive SJIA. Overrepresentation analysis of the overlapping 80 DIRAGs resulted in five pathways being overrepresented: “Phase 0—rapid depolarization”, “Protein methylation”, “Netrin-1 signaling”, “RMTs methylate histone arginines”, and “Vesicle-mediated transport” (*p* ≤ 0.05; not adjusted) (Appendix A). “Phase 0—rapid depolarization” has the highest enrichment ratio of 17.3, where two out of eight proteins are significant in paired and unpaired active versus inactive SJIA analysis (calmodulin 3 (CALM3) and fibroblast growth factor 13 (FGF13)). The highest number of matching proteins between the pathway and the resulting DIRAG list is for the pathway “Vesicle-mediated transport”, where 20 out of 350 proteins match.

#### 3.2.2. Active SJIA versus Healthy Controls

The comparison between active SJIA patients (n = 26) and controls (n = 23) resulted in 2118 significant (*p* ≤ 0.05) DIRAGs, of which 71 pass adjusting for multiple testing (adj.p ≤ 0.05) (see a in Table 2, Figure 3; for a full list, see Appendix A). In total, 945 DIRAGs are more reactive in active SJIA patients, and 1173 are more reactive in inactive SJIA patients. The 2118 significant DIRAGs cover 1719 unique antigens. The top 5 most reactive DIRAGs in active SJIA are MAFF (log2FC = 1.04; adj.p = 0.02), hypoxanthine phosphoribosyltransferase 1 (HPRT1; log2FC = 0.96; adj.p = 0.006), elongation factor Tu GTP binding domain containing 2 (EFTUD2; log2FC = 0.93, *p* = 0.01), proteasome 26S subunit, ATPase 4 (PSMC4; log2FC = 0.92; adj.p = 0.04), and protein-L-isoaspartate (D-aspartate) O-methyltransferase (PCMT1; log2FC = 0.9; *p* = 0.001). The top 5 least reactive DIRAGs in active SJIA are cilia and flagella-associated protein 36 (CFAP36; log2FC = −1.15; *p* = 0.04), sorting nexin 1 (SNX1; log2FC = −1.13; adj.p = 4 × 10^−6^), GRB2-associated regulator of MAPK1 subtype 1 (GAREM1; log2FC = −1.11; *p* = 0.003), ERCC excision repair 5, endonuclease (ERCC5; log2FC = −1.1; *p* = 0.006), and spectrin beta, non-erythrocytic 1 (SPTBN1; log2FC = −1.1; *p* = 0.003) (see Appendix A).

Analyzing the pathways using the resulting DIRAGs from the comparison between active SJIA versus healthy controls shows no significant enriched pathway but an involvement of the innate immune system. Two pathway groups are enriched in inactive SJIA (not significant): “Cytokine Signaling in Immune System” and “Immune System”. Those enriched in active disease (not significant) include “Eukaryotic Translation Initiation”, “Synthesis of DNA”, “Senescenece-Associated Secretory Phenotype (SASP)”, “Metabolism of Nucelotides”, “Ub-specific Processing Proteases”, “Signaling by ROBO Receptors”, “Transport of Small Molecules”, “Host Interactions of HIV factors” (See Appendix A).

#### 3.2.3. Inactive SJIA versus Healthy Controls

The last comparison including inactive SJIA samples (n = 40), and healthy controls (n = 23) showed 2317 significant DIRAGs (1879 unique antigens); of those, 138 (132 unique) passed adjusting for multiple testing (see b in Table 2, Figure 3; for a full list, see Appendix A). In inactive SJIA patients, 1169 DIRAGs are more reactive in inactive SJIA and 1148 more reactive in healthy controls. The log2FC ranges from 1.07 (HPRT1) to −1.14 (PSMD4). The top 5 most reactive DIRAGs in inactive disease are HPRT1 (log2FC = 1.067; adj.p = 1.5 × 10^−4^), lipase maturation factor 2 (LMF2; log2FC = 0.93; adj.p = 0.016), cathepsin C (CTSC; log2FC = 0.92; adj.p = 0.003), ribosomal protein S11 (RPS11; log2FC = 0.89; adj.p = 0.03), and F-box and WD repeat domain containing 5 (FBXW5; log2FC = 0.89; *p* = 0.002). The top 5 most reactive DIRAGs in healthy controls are FKBPL (log2FC = −1.4; *p* = 0.006), two proteins of proteasome 26S subunit ubiquitin receptor, non-ATPase 4 (PSMD4; log2FC = −1.14/−1.1; *p* = 0.04 (adj)/0.001), SNX1 (log2FC = −1.1; adj.p = 1.1 × 10^−6^), and GAREM1 (log2FC = −1.1; *p* = 0.002) (see Appendix A).

GSEA with the 2317 significant DIRAGs of the group comparison between inactive SJIA samples and healthy controls shows 14 significantly enriched pathways in inactive disease, which can be summarized into seven pathway groups with an FDR ≤ 0.05: “Selenoamino acid metabolism”, “Phospholipid metabolism”, “SRP-dependent cotranslational protein targeting to membrane”, “Nonsense-Mediated Decay (NMD) independent of the Exon Junction Complex (EJC)”, “Eukaryotic Translation Elongation”, “L13a-mediated translational silencing of ceruloplasmin expression”, and “Translation” (See Appendix A). There are no significant pathways that are enriched in the healthy control group.

#### 3.2.4. Overlapping DIRAGs in All Three Group Comparisons

In total, three significant DIRAGs (*p* ≤ 0.05, not adjusted) are common in all group comparisons (SJIA active versus inactive, paired + unpaired, SJIA inactive versus controls, and SJIA active versus controls), MAFF, CTSC, and eukaryotic translation elongation factor 1 beta 2 (EEF1B2) (see Appendix A). The log2FC of the overlapping DIRAG ranges from MAFF (log2FC = 1.04), comparing active SJIA to healthy controls, to EEF1B2 (log2FC = −0.6) comparing inactive SJIA to healthy controls (see Appendix A).

The mean reactivity against MAFF has the highest reactivity in the active SJIA group (8.9), which is followed by the inactive SJIA group (8.3) and healthy controls (7.8). EEF1B2 has the highest reactivity in the control group (7.6); active SJIA (7.3) is higher than inactive SJIA (7.0). Differences in CTSC show the highest reactivity in the inactive SJIA (9.3) group, which is followed by the active SJIA (8.9) group, and the healthy controls (8.4) have the lowest reactivity.

## 4. Discussion

Here, we present the antibody reactivities of patients with systemic juvenile idiopathic arthritis (SJIA). To identify differentially reactive autoantigens, we analyzed 110 16k protein microarrays comprising 15312 full-length *E.coli* cDNA expression clones from three different sample groups. The sample groups are active SJIA, inactive SJIA, and healthy controls. Paired samples with both active and inactive disease state were available for 18 patients (see Table 1).

The overall IgG concentration after IgG extraction from serum between active and inactive SJIA compared to the healthy controls did not differ significantly. Comparing the IgG concentrations of paired active and inactive SJIA patients, the group means showed a significant (*p* = 0.02) difference of 0.34 mg/mL. In a study by Rayhan et al. [45], the serum IgG concentrations during active and inactive disease states in 33 patients with juvenile idiopathic arthritis were compared. They showed that the IgG concentration is comparable to healthy controls in the inactive state but is abnormal in the active disease state, which follows our findings.

Comparing SJIA patient pairs in their active state versus inactive disease shows a higher antibody reactivity to the transcription factor MAFF (v-maf musculoaponeurotic fibrosarcoma oncogene homolog F) in active disease both in the paired and unpaired comparison (see c,d in Table 2, Figure 3). Comparing active SJIA versus healthy controls also shows a higher reactivity against MAFF in active SJIA (see a in Table 2). Comparing all three analysis groups shows that the sero-reactivity of MAFF significantly decreases between active SJIA, inactive SJIA, and healthy control samples (see Appendix A). Moon et al. [46] showed that an overexpression of MAFF leads to an activation of the signal transducer and activator of transcription 3 (STAT3) pathway, which signals upstream of IL-6, a known biomarker and therapeutic target in SJIA [47]. A study by Milner et al. [48] identified gain-of-function mutations of STAT3 in whole-exome sequencing data of patients with childhood-onset autoimmunity and lymphoproliferative disease without a genetic cause, hypothesizing that mutations in STAT3 cause altered regulatory T cells and cytokine signaling, causing the immunomic reaction. A study by Li et al. [49], investigating the effect of microRNA (miR)-19a, miR-21 on the JAK/STAT signaling pathway, showed that STAT3 is highly expressed in SJIA patients. MAFF could activate the STAT3 pathway, leading to an overproduction of cytokines; with an overproduction of IL-6, the STAT3 pathway is deactivated. This circle of inflammation could result in the recurrent fever attacks in SJIA.

During our analysis, three of eight PLCG1 (phospholipase C gamma 1) proteins present on the 16k protein microarray showed a significantly lower antibody reactivity in active disease compared to inactive disease in a paired analysis (see c in Table 2, Figure 3). Unpaired, two of eight PLCG1 proteins showed a significantly lower reactivity in active disease within the top 10 DIRAGs (see d in Table 2). The protein PLCG1 catalyzes the hydrolysis of phosphatidylinositol 4,5-bisphosphate to generate the second messenger molecules, inositol 1,4,5-trisphosphate (IP3, or InsP3) and diacylglycerol (DAG) [50]. Interestingly, the dysregulation of phospholipase C γ 2 (PLCG2) is known to result in autoinflammatory diseases like Phospholipase Cγ-Associated Antibody Deficiency and Immune Dysregulation (PLAID) [51] and autoinflammation and PLAID (APLAID) [52]. Dysregulations in PLCG1 were mostly associated with cancer [53,54]. A study by Fu et al. [55] showed that PLCG1-deficient mice develop inflammatory/autoimmune disease. Both PLCG1 and PLCG2 belong to the PLCG enzyme family but have distinct characteristics and functions. They have different roles in cellular signaling, where PLCG1 is involved in the signaling pathways triggered by receptor tyrosine kinases and PLCG2 is predominantly associated with the signaling pathway mediated by immune receptors [56]. The role of PLCG1 in SJIA needs to be further investigated.

When comparing inactive SJIA patients to healthy controls, antibodies against cathepsin C (CTSC) are significantly more reactive in inactive disease compared to healthy controls (b in Table 2, Figure 3). In a study by Alam et al. [57], treating peritoneal macrophages and the mouse macrophage cell line with LPS and an active monomer of CTSC, it was shown that upregulated CTSC could induce macrophage activation toward the M1 phenotype, leading to a self-activation mechanism, and further leading to a circle in chronic inflammation. This indicates that lower antibody reactivity against CTSC in active SJIA patients, resulting in an upregulation of CTSC, could contribute to chronic inflammation by macrophage activation. CTSC is significant in all group comparisons analyzed in the present study, with the highest reactivity in inactive SJIA, followed by active SJIA, while healthy controls have the lowest reactivity against CTSC (Figure 3).

In the present study, we identified DIRAGs that affect the Nuclear Factor-kappa Binding protein (NF-κB) pathway. NF-κB is an important regulator of immune and inflammatory responses. Activated NF-κB expresses genes that encode cytokines (TNF, IL-1, IL-6), and in the innate response, it facilitates T-cell activation [58]. A dysregulated NF-κB pathway could lead to inflammation in SJIA patients.

Comparing the antigenic reactivity of unpaired active versus inactive SJIA showed that active SJIA have a higher reactivity against FK506-Binding Protein like (FKBPL) than inactive SJIA (d in Table 2, Figure 3), and healthy controls have a significantly higher reactivity compared to inactive SJIA (b in Table 2, Figure 3), showing that active SJIA have a similar level of antigenic reactivity against FKBPL than healthy controls. FKBPL is described as targeting the inflammatory STAT3 pathway [59], and it is also a negative regulator of NF-κB activation [60], hence lowering inflammatory processes. This is contrary to our results, where inactive SJIA patients have the lowest antigenic reactivity against FKBPL compared to active SJIA and healthy controls.

It is also shown that abnormal or knockout LMNA (lamin A/C) can activate the NF-κB-mediated inflammatory response [61]. In our study, there was higher antibody reactivity against LMNA in inactive disease compared to active SJIA (b in Table 2, Figure 3), leading to an inactivated NF-κB pathway in inactive SJIA.

In our study, FBXW5 (F-box and WD repeat domain containing 5) was significantly more reactive in inactive SJIA samples and active SJIA samples compared to healthy controls (a,b in Table 2, Figure 3). It was reported that dysregulated FBXW5 expression could activate the NF-κB pathway, further releasing pro-inflammatory cytokines [62]. Another study showed that FBXW5 knockdown in spinal cord injury mice could repress inflammation and improve spinal cord injury development [63]. Interestingly, both active and inactive SJIA groups have a similar reactivity against FBXW5; hence, the reactivity is independent from the activity status, possibly leading to a constant slightly increased level of cytokines.

Comparing active SJIA to healthy controls shows that SPTBN1 has a significantly lower reactivity in active SJIA (a in Table 2). SPTBN1 is an actin crosslinking and molecular scaffold protein. A study by Lin et al. [64] analyzing the effect of SPTBN1 in HCC cell lines and liver tissue on expression of pro-inflammatory cytokines concluded that the loss of SPTBN1 in a hepatocellular carcinoma cell line enhanced the expression of IL-1a, IL-1b, and IL-6, and it further activates the NF-κB pathway. The same was shown by Dai et al. [65] analyzing the effect of SPTBN1 in rheumatoid arthritis NC-fibroblast-like synoviocytes. The lower reactivity of SPTBN1 in active SJIA identified in this study could enhance the expression of IL-1a, IL-1b, and IL-6 in SJIA the same way as described in the studies above.

Another antigen within the top 5 in active SJIA compared to healthy controls involved in inflammatory processes is elongation factor Tu GTP binding domain containing 2 (EFTUD2) [66,67,68]. Our study showed that the reactivity against EFTUD2 increases in active disease compared to healthy controls (a in Table 2). A comparative RNAi screen by De Arras et al. [66] identified EFTUD2 as a novel regulator of the innate immunity in *C. elegans*, hypothesizing that the conserved innate immune regulatory function in *C. elegans* will likely also affect humans. Taylor et al. [67] showed that rs9910936, most likely an SNP in EFTUD2, is within the top 25 associated candidate genes when analyzing the response of methrotrexate in early rheumatoid arthritis patients by a genome-wide association study (GWAS). Another study, analyzing the global expression of splice variants to identify channel-dependent signaling mechanisms, identified an association between the activation of human macrophage SCN5A and SCN10A with an increase in EFTUD2, hence linking innate immune signaling in human macrophages to DNA repair [68]. In active SJIA patients, an increased reactivity against EFTUD2 could contribute to chronic inflammation by macrophage activation.

As with EFTUD2, PCMT1 is more reactive in active SJIA compared to healthy disease (a in Table 2, Figure 3). PCMT1 is a member of the type 2 class of protein carboxyl methyltransferase enzymes. Studies show an association of elevated PCMT1 levels with various cancer types like breast cancer [69,70,71,72,73,74]. Another study constructing a genetic network for rheumatoid arthritis by evaluating genome-wide SNPs showed that PCMT1 is within the 41 identified significant SNPs, which is relevant to rheumatoid arthritis [75].

To summarize the resulting DIRAGs in biological pathways, reactome GSEA with resulting DIRAG lists has been performed. GSEA from the DIRAG list comparing active SJIA versus inactive SJIA showed that the pathway “Regulation of expression of SLITs and ROBOs”, enriched in inactive disease, had the highest negative normalized enrichment score, where 35 of 51 antigens were present in the resulting DIRAG list, and they are all more reactive in inactive disease. Slits (slit guidance ligands) and Robos (Roundabout receptors) play an important role in muscle cell formation, cell migration, stem cell growth, angiogenesis, organ development, and tumor formation. Their downstream targets impact the cell mobility, including kinases comprising Hakai and Myo9b and GTPases containing the Rho-family including RhoA [76,77]. RhoA is known as a key regulator of Innate and Adaptive immunity [78,79]. Antigenic reactivity could lead to the downregulation of this pathway followed by a dysregulation of RhoA, and therefore, it could activate inflammation in SJIA.

Another pathway significantly enriched in inactive disease compared to active disease is “Nonsense-Mediated Decay (NMD)” (24 of 35 antigens affected). This pathway acts as a quality control pathway as it degrades some mRNAs bearing premature termination codons (PTCs). A dysregulated NMD in cancer can lead to the inactivation of tumor-suppressor genes and an increased cancer cell growth [80]. A study by Grandemange et al. [81], analyzing the post-transcriptional regulation of the Mediterranean fever gene by NMD, showed that several protein isoforms could exist in leukocytes with different sub-cellular localizations and inflammatory functions. Therefore, it could be partly responsible for the clinical heterogeneity in FMF patients [81]. It was also shown that UPF1, a canonical NMD factor, and Regnase-1 can regulate the early phase of inflammation response by degrading cytokine mRNAs [82]. Our findings show that the NMD pathway is enriched in inactive SJIA patients, indicating that this pathway could suppress inflammation in inactive SJIA.

GSEA of the resulting DIRAGs between inactive SJIA and healthy controls showed seven significantly enriched pathways in inactive disease. “Selenoamino acid metabolism” is the top enriched GSEA pathway comparing inactive SJIA to healthy controls, with 37 proteins of 48 significantly more reactive in inactive SJIA. In this pathway, sulfur is substituted by selenoamino acids. The literature research showed one SJIA case with lower plasma selenium levels compared to healthy peers analyzed in 1986 [83]. Generally, selenium regulates inflammatory and immune responses as it induces higher levels of inflammatory cytokines and is associated with rheumatoid arthritis and inflammatory bowel disease [84,85,86].

## 5. Conclusions

In conclusion, we identified several differentially reactive antigens involved in inflammatory processes. Intriguingly, many of the identified antigens had a high reactivity against proteins involved in the NF-κB pathway, and it is also notable that many of the detected DIRAGs are described as dysregulated in rheumatoid arthritis. Our data highlight novel proteins and pathways potentially dysregulated in SJIA and offer a unique approach at unraveling the underlying disease pathogenesis in this chronic arthropathy. Although the data need to be validated with an independent sample cohort, the results indicate a previously unrecognized and significant involvement of (auto)antibodies in systemic juvenile idiopathic arthritis.

## Figures and Tables

**Figure 1 biomolecules-13-01392-f001:**
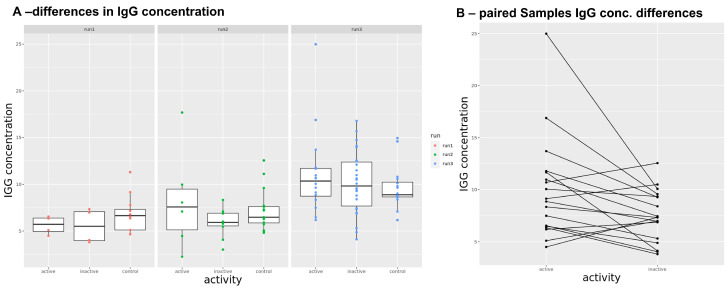
(**A**) Differences in IgG concentrations between the three analysis days. The concentration of run 3 is higher compared to the similar concentration of run 1 and run 2. To overcome this difference, IgG was diluted to a standardized concentration of 0.2 mg/mL and put on the microarray. A comparison of IgG concentrations of active and inactive SJIA to healthy controls shows no significant difference for each of the three analysis days. (**B**) Differences in IgG concentration of paired SJIA samples were significant (*p* = 0.02) with a difference in mean of 2.5 mg/mL, where active SJIA samples have a higher concentration (mean = 9.98 mg/mL) than inactive SJIA samples (mean = 7.49 mg/mL).

**Figure 2 biomolecules-13-01392-f002:**
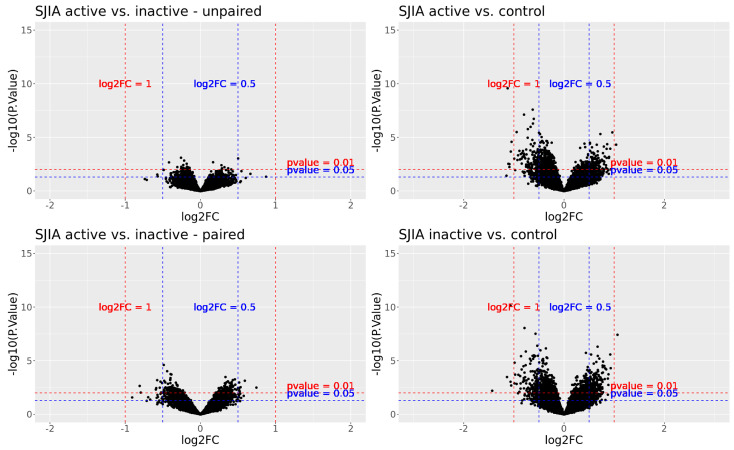
Volcano plot of the resulting DIRAGs from four different group comparisons.

**Figure 3 biomolecules-13-01392-f003:**
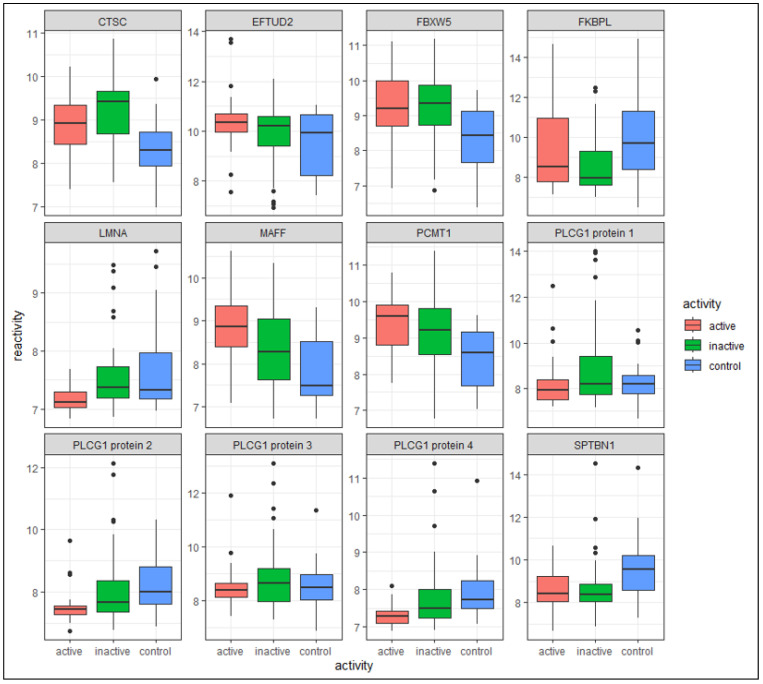
Differentially reactive antigens involved in inflammatory processes resulting from group comparisons between all sample groups.

**Table 1 biomolecules-13-01392-t001:** Patient characteristics: signs and symptoms of active patients. The % documented value indicates the number of patients where this information is documented.

	SJIA	Active	Inactive	Healthy Controls
**n**	66	26 (39.4%)	40 (60.6%)	23
**Sex (% male)**	35 (53)	14 (53.8)	21 (52.5)	12 (52.2)
**Age at diagnosis ^1,2^**	7.1 (0.6–17.4)	6.6 (0.6–17.3)	7.1 (0.8–17.4)	n/a
**Age at sampling ^1,2^**	11.5 (1.0–20.8)	10.0 (1.0–20.2)	12.6 (2.4–20.8)	14.7 (10.5–15.9)
**Disease duration ^1,2^**	3.5 (<0.1–17.3)	1.6 (<0.1–15.3)	3.4 (0.4–17.3)	n/a
**CRP (mg/dL) ^2^**		6.8 (0.5–21.1) % documented: 92.3	<0.5 % documented: 95	<0.5 % documented: 95.7
**ESR (mm/hour) ^2^**		66 (10–134) % documented: 80.8	5.1 (1–30) % documented: 92.5	3 (<1–13) % documented: 52.2
**Leukocytes/nL ^2^**		15.2 (5–23.8) % documented: 84.6	5.9 (3.6–9.4) % documented: 92.5	5.5 (<1–7.9) % documented: 95.7
**S100A12 (ng/mL) ^2^**		1195 (180–19,410) % documented: 100	39 (8–2450) % documented: 100	n/a
**S100A8/A9 (ng/mL) ^2^**		7320 (720–220,850) % documented: 34.6	560 (100–20,710) % documented: 27.5	n/a
**Steroids ^3^**		18 (69.2)	9 (22.5)	0
**bDMARDs ^3^**		21 (80.8)	24 (60)	0
**csDMARDs ^3^**		10 (38.4)	14 (35)	0

^1^ in years, ^2^ median (range), ^3^ n(%).

**Table 2 biomolecules-13-01392-t002:** Top 10 most and least reactive antigens resulting from group comparisons between (a) active SJIA vs. control, (b) inactive SJIA vs. control, (c) active SJIA vs. inactive SJIA—paired, and (d) active SJIA vs. inactive SJIA—unpaired.

	Higher in	log2FC	FC	*p* Value	Gene		Higher in	log2FC	FC	*p* Value	Gene
a—active vs. control	active	1.039	2.055	0.000	MAFF	b—inactive vs. control	inactive	1.067	2.095	0.000	HPRT1
		0.964	1.951	0.000	HPRT1			0.931	1.907	0.000	LMF2
		0.931	1.907	0.010	EFTUD2			0.923	1.896	0.000	CTSC
		0.923	1.895	0.000	PSMC4			0.887	1.849	0.000	RPS11
		0.898	1.864	0.001	PCMT1			0.885	1.847	0.002	FBXW5
		0.895	1.860	0.001	ADK			0.875	1.834	0.015	EMC10
		0.892	1.855	0.025	EFCC1			0.875	1.834	0.000	RPS27A
		0.880	1.841	0.012	FMNL1			0.841	1.792	0.000	TFAP4
		0.866	1.822	0.009	ISYNA1			0.827	1.774	0.046	BAG5
		0.861	1.816	0.000	HDAC2			0.820	1.765	0.000	AXIN2
	control	−1.147	0.452	0.038	CFAP36		control	−1.433	0.370	0.006	FKBPL
		−1.125	0.458	0.000	SNX1			−1.137	0.455	0.000	PSMD4
		−1.111	0.463	0.003	GAREM1			−1.067	0.477	0.001	PSMD4
		−1.090	0.470	0.006	ERCC5			−1.066	0.478	0.000	SNX1
		−1.088	0.470	0.003	SPTBN1			−1.054	0.482	0.002	GAREM1
		−1.062	0.479	0.000	MZF1			−0.990	0.503	0.001	OTUB1
		−1.047	0.484	0.000	ASCC2			−0.987	0.505	0.000	ASCC2
		−0.994	0.502	0.001	KIF2C			−0.975	0.509	0.000	MZF1
		−0.975	0.509	0.011	MLLT6			−0.946	0.519	0.001	PSMD4
		−0.946	0.519	0.000	CALB2			−0.917	0.530	0.011	ERCC5
c—active vs. Inactive (paired)	active	0.745	1.676	0.003	IARS2	d—active vs. Inactive (unpaired)	active	0.873	1.832	0.047	FKBPL
		0.591	1.507	0.001	CALM3			0.665	1.586	0.025	ARFIP2
		0.576	1.491	0.019	MEIS3			0.546	1.460	0.014	MAFF
		0.542	1.456	0.030	ZNF232			0.502	1.416	0.001	CALM3
		0.535	1.449	0.003	MAFF			0.473	1.388	0.033	TMEM183A
		0.530	1.444	0.012	IGHV3−21			0.444	1.360	0.047	RTN4
		0.523	1.437	0.008	BYSL			0.443	1.359	0.034	APBA2
		0.518	1.432	0.004	SDHA			0.430	1.348	0.015	PPP1CA
		0.516	1.430	0.006	ARID1A			0.426	1.344	0.030	SAP30BP
		0.516	1.430	0.001	TPM4			0.422	1.340	0.046	ACLY
	inactive	−0.907	0.533	0.027	PLCG1		inactive	−0.573	0.672	0.030	PLCG1
		−0.807	0.572	0.002	AGAP2			−0.568	0.675	0.044	AGAP2
		−0.792	0.577	0.009	APBA2			−0.491	0.711	0.011	PLCG1
		−0.694	0.618	0.026	PLCG1			−0.487	0.713	0.011	CLDN3
		−0.667	0.630	0.042	PLCG1			−0.417	0.749	0.002	LMNA
		−0.592	0.663	0.005	DOCK2			−0.394	0.761	0.023	BUB1B
		−0.587	0.666	0.004	RSL1D1			−0.394	0.761	0.033	CTSC
		−0.581	0.669	0.029	PLCG1			−0.388	0.764	0.046	ACAD10
		−0.574	0.672	0.001	RPL10A			−0.387	0.765	0.022	RPL36A
		−0.570	0.673	0.015	FLAD1			−0.372	0.773	0.048	SMARCB1

## Data Availability

The data presented in this study are available on request from the corresponding author.

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
