# Peer review of "Patients with Systemic Juvenile Idiopathic Arthritis (SJIA) Show Differences in Autoantibody Signatures Based on Disease Activity"

_biomolecules, 2023, doi:10.3390/biom13091392_

Round 1

Reviewer 1 Report

Krainer et al uses protein arrays to show that antibodies reacting against several recombinant proteins can be detected in the sera of systemic JIA patients. It is a study in an area that has rarely been explored, and may provide important insights to the understanding of sJIA pathology. As such, this is a study of interest to the field.

However, unless the authors mean something very specific by ‘autoantibody signatures in SJIA’ (line 55), the authors must mention previous studies showing autoantibodies in sJIA. For example, Hugle et al, 2014 DOI: 10.1186/1546-0096-12-28 describes antinuclear antibodies (ANA) and rheumatoid factor (RF) in sJIA patients. Other papers: Schulz et al, 2022, Serra et al, 1999 (Clinical and Experimental Rheumatology 1999; 17: 375-380), and Bloom et al, 2007, Rheumatol Int. 2007;27:655–660.

Other points:

What was the criteria to determine that a patient was ‘active’ or ‘inactive’?

Supplementary Table 1 is not clear. For example, for Arthritis, accordingly to the label, 5 patients have arthritis, and the percentage is 62.5? And so on for the other categories. What does it mean ‘% documented :31’ in all categories? I cannot see what the number 1 at the bottom of the Table refers to.

Lines 64 and 65 says that ‘For 18 patients, paired samples during active and inactive fever periods were taken.’, but Supplementary Table 1 shows 4 active patients with fever. Please clarify.

I recommend to move the information from Supplementary Table 1 into Table 1 in the main paper.

Line 198: ‘In total 66 SJIA patients classified by ILAR criteria’ – please move this information to the Material and methods

Legend of Fig 1: ‘To overcome this difference, an adjusted amount of IgG was put on the microarray.’ – should be in the text, and please clarify how the adjustment was made.

I am curious if the authors analyzed any potential effect of medication on the results.

Line 200: ‘To compare the immunomic differences, active and inactive disease states were compared to the healthy controls individually. I suggest changing ‘individually’ to ‘separately’.

Author Response

I want to thank the reviewers for their quick, constructive comments and suggestions. All raised issues are addressed in the response to reviewer document where you can find the point-by-point responses to the reviewers’ comments.

The changes target diagnosis criteria of SJIA patients, Table 1, corrections of the IgG levels, and wording throughout the manuscript.

Reviewer 2 Report

Dear Authors and Editors!

Thank you for the opportunity to review the manuscript.

In the study Authors analyzed the immunological responses of patients with SJIA, using human proteome arrays presenting immobilized recombinantly expressed human proteins, to analyze the involvement of autoantibodies  in SJIA. It could be used in the differentiating between autoimmune and antiinflammatory diseases.

I have several questions and suggestions:

1) The results of studied population are in the Materials and duplicated in the Table. I suggest to place them in the results and remain only principal differences without digits.

2) Why active sJIA had so small concentration of CRP?

3) What does it mean % documented? This is unclear.

4) What kind of Ig G did Authors measure? The level is approx 1 mg/ml is a very small for common IgG level? Did the treatment affect this?

5) What kind of autoantibodies did Author measure. This is unclear.

The manuscript difficult for reading and Introduction does not contain information about the further findings. It contains only common clinical information about sJIA. Please elaborate the introduction to be clear why you perform this study with this type of analysis. Will be good if discussion interferes the main points of the Introduction.

Author Response

(The authors gave the same response as above.)

Round 2

Reviewer 2 Report

Dear Authors!

Thank you for submission the revised version.

I have no any question.

Good luck!